# Whole-Genome Sequencing of Human Enteroviruses from Clinical Samples by Nanopore Direct RNA Sequencing

**DOI:** 10.3390/v12080841

**Published:** 2020-07-31

**Authors:** Carole Grädel, Miguel A. Terrazos Miani, Christian Baumann, Maria Teresa Barbani, Stefan Neuenschwander, Stephen L. Leib, Franziska Suter-Riniker, Alban Ramette

**Affiliations:** 1Institute for Infectious Diseases, University of Bern, 3001 Bern, Switzerland; carole.graedel@ifik.unibe.ch (C.G.); miguel.terrazos@ifik.unibe.ch (M.A.T.M.); christian.baumann@ifik.unibe.ch (C.B.); mariateresa.barbani@ifik.unibe.ch (M.T.B.); stefan.neuenschwander@ifik.unibe.ch (S.N.); stephen.leib@ifik.unibe.ch (S.L.L.); franziska.suter@ifik.unibe.ch (F.S.-R.); 2Graduate School for Cellular and Biomedical Sciences, University of Bern, 3012 Bern, Switzerland

**Keywords:** enterovirus, direct RNA sequencing, nanopore

## Abstract

Enteroviruses are small RNA viruses that affect millions of people each year by causing an important burden of disease with a broad spectrum of symptoms. In routine diagnostic laboratories, enteroviruses are identified by PCR-based methods, often combined with partial sequencing for genotyping. In this proof-of-principle study, we assessed direct RNA sequencing (DRS) using nanopore sequencing technology for fast whole-genome sequencing of viruses directly from clinical samples. The approach was complemented by sequencing the corresponding viral cDNA via Illumina MiSeq sequencing. DRS of total RNA extracted from three different enterovirus-positive stool samples produced long RNA fragments, covering between 59% and 99.6% of the most similar reference genome sequences. The identification of the enterovirus sequences in the samples was confirmed by short-read cDNA sequencing. Sequence identity between DRS and Illumina MiSeq enterovirus consensus sequences ranged between 94% and 97%. Here, we show that nanopore DRS can be used to correctly identify enterovirus genotypes from patient stool samples with high viral load and that the approach also provides rich metatranscriptomic information on sample composition for all life domains.

## 1. Introduction

Enteroviruses (EVs) present a major burden for human health and health care systems worldwide, with large outbreaks consisting of hundreds of thousands of hospitalized cases occurring periodically [1]. EVs are associated with a wide variety of symptoms, ranging from mild respiratory diseases to severe neurological infections, leading potentially to death [2]. These single-stranded RNA viruses that belong to the *Picornaviridae* family possess a relatively small genome size ranging from 7.2 to 8.5 kb. For routine diagnostics, EV presence is generally determined by real-time, reverse-transcription PCR (qRT-PCR) assays, complemented by partial sequencing of the VP1–VP4 coding regions for genotyping [2,3]. Due to the lack of proofreading mechanism in RNA replication, EV genomes are highly variable and the likely subject of within- and between-genome recombinations [4,5]. Therefore, when diagnostic tests solely rely on PCR using conserved primer sequences, false-negative results cannot be excluded. There is thus a need to generalize the assessment of EV diversity and evolution via a whole-genome approach, rather than from the limited information gained from sequencing single genomic regions.

The main limitation towards a general adoption of whole-genome sequencing (WGS) for better EV characterization and better understanding of their evolution and epidemiology is mostly of a technological nature: WGS approaches are more expensive, technically demanding and hence time consuming than standard molecular assays. As such, they are generally not routinely used in diagnostic laboratories. In the case of RNA viruses, the challenge is even more exacerbated because various molecular steps (RNA purification, extraction, cDNA synthesis, and optionally amplification) add to the turnaround time and final cost of the assays. Clinical WGS applications are mostly based on Sanger sequencing and on second-generation sequencing platforms, with Illumina MiSeq/HiSeq (Illumina, San Diego, CA, USA) and Ion Torrent machines leading the market, as benchtop sequencing technologies enable high-throughput sequencing at an affordable price per sample when samples are multiplexed [6]. Third-generation sequencers, i.e., SMRT technology (Pacific Biosciences, Menlo Park, CA, USA) and nanopore sequencing (Oxford Nanopore Technologies (ONT), Oxford, UK), have recently emerged as complement to or replacement of second-generation sequencers, by enabling the sequencing of single, long DNA molecules, a feature that is particularly attractive in the context of sequencing full-length viral genomes [7].

Nanopore sequencing technology has been successfully applied to genome sequencing of RNA viruses. Based on viruses propagated on cell lines, viral transcriptomes have been examined by nanopore sequencing, for instance for porcine circovirus [8], herpes simplex virus [9], hepatitis C virus [10], cultured influenza virus A, and human cytomegalovirus (HCMV) [11]. Additionally, nanopore sequencing has been used successfully to sequence whole EV genomes from viral cDNA extracted from cell cultures: in their proof-of-concept study [12], the authors demonstrated that nanopore sequencing could be applied for rapid routine whole-genome sequencing of EV with sufficient accuracy compared to Sanger sequencing. Metagenomic nanopore sequencing of influenza virus was performed directly from randomly amplified viral cDNA obtained from clinical respiratory samples [13]. Furthermore, ONT has presented a new, potentially revolutionary nanopore sequencing application, which allows sequencing of RNA molecules directly, i.e., without the pre-requirement of cDNA synthesis or PCR amplification [14]. This method, direct RNA sequencing (DRS), yields full-length, strand-specific RNA sequences and enables the direct detection of nucleotide modifications in native RNA molecules. DRS has been used in several studies, including human poly(A) transcriptome [15] and DNA virus transcriptome, such as HSV herpes simplex virus type 1 (HSV-1) during productive infection of primary cell [16]. DRS was also used for sequencing RNA genomes of, e.g., Pseudorabies virus propagated on immortalized porcine kidney epithelial cell line [17], influenza A virus (H1N1) from infected chicken eggs [18], human Coronaviruses viral RNAs produced in cell cultures [19], and many other examples of complete sequencing of multiple, single-stranded RNA (ssRNA) viruses obtained with or without poly(A)-tailing of RNA viruses obtained from cell cultures [20].

Here, in a proof-of-concept study, we apply nanopore DRS to EV-positive stool samples and show that whole RNA genomes of enteroviruses can be retrieved with enough genomic information for the characterization of the infectious agents. We further analyze the metatranscriptomic data provided by the DRS approach and compare it with that obtained by Illumina MiSeq sequencing of the same samples.

## 2. Materials and Methods

### 2.1. Sample Description

In this study, we used three independent patient stool samples (E590, E372, E026), which were sent for enterovirus diagnostics to the clinical diagnostic laboratory of the Institute for Infection Diseases, Bern, Switzerland. The samples were collected in 2017 (E590, E372) and 2018 (E026) and had a cycle threshold (Ct) value of 23.0 (E590), 18.8 (E372) and 22.5 (E026) via real-time PCR using enterovirus-specific primers (see description below). Ethics approval was granted on 8 March 2018 by the Swiss Ethics committee on research involving humans to conduct sequencing of enteroviruses in clinical samples stored in the IFIK biobank (BASEC-Nr: Req-2018-00158).

### 2.2. Routine Diagnostic Approach

After homogenization with a sterile pipette, approximately 0.5 g of stool sample was added to 4 mL of transport medium [21] containing 5–10 glass beads (diameter 2 mm; Merck AG, Zug, Switzerland). After 30 s of vortexing, the suspension was centrifuged for 5 min at 3350 g and the resulting supernatant was filtered through a pore size of 0.2 µm with 20% penicillin/streptomycin (Biochrom, Berlin, Germany). This preparation was further extracted on the EasyMAG platform (bioMérieux, Geneva, Switzerland) for real-time PCR and with TRIzol LS Reagent (Thermo Fisher Scientific, Reinach, Switzerland) for Illumina MiSeq sequencing (Figure 1).

### 2.3. RNA Extraction and Purification

A total of 750 µL of TRIzol LS Reagent (Thermo Fisher Scientific) was added per 250 µL of stool suspension in PBS. Following homogenization by pipetting, the sample was transferred to Phasemaker tubes (Thermo Fisher Scientific). After incubation for 5 min, 200 µL of chloroform was added, and the tubes were shaken vigorously by hand for 15 s. After 15 min incubation, the sample was centrifuged for 5 min at 16,000× *g*, at 4 °C. The aqueous phase was transferred to a new tube and 10 µg of carrier RNase-free glycogen (Thermo Fisher Scientific) was added, followed by 500 µL isopropanol. After incubation for 10 min, the sample was centrifuged for 10 min at 12,000× *g*, at 4 °C, and the supernatant was discarded. The total RNA precipitate was resuspended in 1 mL of 75% ethanol. The sample was vortexed briefly, and then centrifuged for 5 min at 7500× *g*, at 4 °C. The supernatant was discarded and the RNA pellet was air-dried for 10 min, before being resuspended in 20 µL of RNA storage solution (Thermo Fisher Scientific). After incubation at 60 °C for 10 min, the RNA sample was either used directly in downstream applications, or stored at −80 °C. Alternatively, total nucleic acid extraction was performed with NUCLISENS easyMAG from 200 µL of the routine diagnostic stool preparation, to which 2.5 µL carrier RNA (Qiagen AG, Hombrechtikon, Switzerland) was added and eluted in 110 µL. This extract was used for real-time PCR and Sanger genotyping.

### 2.4. Pre-Treatment (Chloroform/Beads/Centrifugation)

We used the WHO-recommended protocol for pre-treating stool samples for enterovirus RNA isolation (Enterovirus surveillance guidelines, [22]) adapted from Nix et al. 2006 [23] as follows: a low amount (0.5 to 1 g) of stool sample was added to PBS (Thermo Fisher Scientific) up to 1 mL volume, to which 0.3 g of glass beads (2 mm diameter, Merck) and 0.5 mL of chloroform (AppliChem, Aesch, Switzerland) were added. The mixture was shaken vigorously using a TissueLyser (Qiagen AG) for 20 min at maximum speed. The suspension was centrifuged at 1500× *g* for 20 min at 4 °C, and approximately 1 mL of the supernatant was transferred to a new 1.5 mL tube and continued with RNA extraction. For DRS of sample E590, RNA was extracted using easyMAG, while for E372 and E026, the TRIzol LS method was used (see Appendix A).

### 2.5. Real-Time RT PCR

One-step RT-PCR was performed with the AgPath-ID one-step kit (Ambion, Reinach, Switzerland) using published primers and probes [24], following the protocol described previously [25]. Primers and probes were synthesized at Microsynth AG, Balgach, Switzerland.

### 2.6. Enterovirus Genotyping

Genotyping of the samples was performed by VP1 amplicon Sanger sequencing as described previously [23,25] and carried out at Microsynth.

### 2.7. Nanopore Sequencing

For nanopore sequencing, we followed the manufacturer’s instructions of the protocol for kit SQK-RNA001 (version DRS_9026_v1_revN_15Dec2016), but substituted Superscript III with Superscript IV (Thermo Fisher Scientific). The input RNA and amounts loaded on flow cells for each experiment are specified in Appendix A. Quantification was performed using a Qubit RNA HS assay kit (RNA) or DNA HS assay (cDNA) kit on a Qubit fluorometer 3.0 (Thermo Fisher Scientific). The reverse-transcribed and adapted RNA was loaded onto R9.4.1 flowcells and sequenced on MinION sequencer. Each DRS run was conducted with a new, previously unused flowcell.

### 2.8. Illumina MiSeq. Primer Design for Coxsackievirus A6 cDNA Synthesis

The following three specific primers for Coxsackievirus A6 were designed for this project to hybridize to Coxsackievirus A6 sequence KJ541158: 2588R_A6: 5′-CCCGTTTCTGCCGCTT-3′ adapted from primer 292 by Oberste et al. [26]; 5672R_A6: 5′-ATATCTCTGAATTTCTCATT-3′ adapted from primer HEV.3C.d1 by Bessaud et al. [27]; EV-3UTR1_A6_rc: 5′-CATATTCACGACCAGATTCCTGGTG-3′ (this study). All primers were synthesized at Microsynth.

### 2.9. cDNA Synthesis for Illumina MiSeq

First-strand synthesis was performed with the SuperScript IV First-Strand Synthesis System as follows: for reverse transcription, a reaction mixture was prepared containing 0.5 µL of the specific primers EV-3UTR1_A6_rc, 2588R_A6, 5672R_A6 (10 µM), or 1 µL of random hexamers (50 µM) or 1 µL of oligo(dT)_20_ (50 µM) together with 1 µL of 10 mM dNTP mix and filled with RNA extract and DEPC-treated water to a total volume of 13 µL. After heating at 65 °C for 5 min and snap cooling, a mixture of 4 µL SSIV Buffer, 1 µL 100 mM DTT, 1 µL ribonuclease inhibitor and 1 µL SuperScript IV reverse transcriptase (200 U/µL) were added. The samples were incubated for 10 min at 50 °C and 10 min at 80 °C (preceded by 10 min at 23 °C when containing random hexamer primers). Subsequently, 1 µL of *E. coli* RNase H (2 U/µL) was added and incubated at 37 °C for 20 min. To this mixture, 10 µL of second-strand synthesis reaction buffer NEBNext (New England Biolabs (NEB), Ipswich, MA, USA), 5 µL NEBNext Second-Strand Synthesis Enzyme Mix and 44 µL of nuclease-free water were added. The reaction mixture was incubated at 16 °C for 60 min. Clean up with 144 µL of Agencourt AMPure XP magnetic beads (Beckman Coulter, Nyon, Switzerland) was performed according to the manufacturer’s instructions, with an elution volume of 35 µL for samples E372 and E026 and 47 µL for sample E590. For the first sample E590, an additional end-prep step was performed, but that step was later removed in the protocol as it was deemed unnecessary: to 45 µL of the elution, 7 µL of Ultra II End-Prep buffer (NEB), 3 µL of Ultra II End-Prep enzyme mix (NEB), and 5 µL of nuclease-free water were added and the mixture was incubated for 5 min at 20 °C and 5 min at 65 °C. After an additional purification step using 60 µL of Agencourt AMPure XP magnetic beads, the cDNA was eluted in 20 µL of nuclease-free water.

### 2.10. Illumina MiSeq Sequencing

Libraries were prepared from unamplified cDNA using Nextera XT DNA Library Prep kit (E590) and Nextera DNA Flex Library Prep kit (E372, E026) (Illumina), and sequenced using an Illumina MiSeq benchtop sequencer generating 2 × 150 bp paired-end reads (v2), according to the manufacturer’s protocols. Sequencing was performed at the Next-Generation Sequencing Platform of the Inselspital, Bern, Switzerland.

### 2.11. Bioinformatic Analysis

Raw FAST5 files produced by MinION sequencing were basecalled under the high accuracy mode using the ONT basecaller Guppy version 3.2.2 with the parameter: “guppy_basecaller --input_path PATH --recursive --save_path PATH --qscore_filtering --min_qscore 7 --flowcell FLO-MIN106 --kit SQK-RNA001 --cpu_threads_per_caller 4 --num_callers 4”. Statistics for nanopore sequencing output are summarized in Appendix A. We did not perform any adapter trimming due to the inaccuracy of the RNA basecaller when calling DNA adapter sequences, which results in unreliable identification [19], but do not expect it to impair our downstream analysis, as basecalled nanopore reads were classified using BLASTN against the NCBI’s nucleotide (*nt*) database (downloaded 03.10.2019), using BLAST version 2.6.0. BLAST results were classified using MEGAN (v. 6.12.3, [28]), with the blast2rma parameters “c false --m 10 --ms 50 --me 0.01 --mpi 0 --top 10 --supp 0.05 --sup 1 --alg naive --mrc 0 --mrefc 0 --ram readCount --a2t nucl_acc2tax-Jul2019.abin”. RNA sequences were mapped to the best-scoring reference genome (whole-genome reference with highest bit-score in BLASTN output) using minimap2 (version 2.5-r601-dirty, [29]). For Illumina MiSeq data (Appendix A)**,** adapter trimming was performed with bbduk.sh from the BBMap package (v 37.80, [30]), and human and rRNA reads were removed by mapping reads against the corresponding databases (GRCh18 genome, cDNA and ncRNA, and SILVA 132 LSU SSU https://www.arb-silva.de/documentation/release-132/) using minimap2 [29]. Non-human, non-rRNA reads were then assembled using SPAdes (parameters: -k 55 --rna --only-assembler; version 3.11.0; [31]). The resulting contigs were subsequently analyzed using BLASTN and MEGAN as described for nanopore data. Sequence identity between Illumina MiSeq and nanopore DRS enterovirus genome consensus sequences was calculated using legacy BLAST 2.2.9. Coverage information was generated using BBMap and plots created using R statistical computing environment (version 3.6.0).

### 2.12. Data Availability

After removal of any human reads, all Illumina MiSeq sequencing data (FASTQ), raw and basecalled ONT data (FAST5 and FASTQ, respectively), and Sanger sequences were deposited in the European Nucleotide Archive (ENA) under the project reference PRJEB38758.

## 3. Results

DRS was performed with enterovirus-positive stool samples with similar viral load (Ct values between 18 and 23) in three independent experiments, consisting of samples from three different patients. The samples were prepared using the WHO-recommended protocol for viral enrichment using chloroform/bead treatment, followed by RNA extraction using TRIzol or easyMAG (Appendix A). We sequenced the total polyadenylated RNA using DRS on a MinION Nanopore sequencer. The runs were continued until there was only negligible sequencing output or until the maximum recommended duration of the run was reached. For validation purposes, all three samples were also subjected to cDNA sequencing using Illumina MiSeq from samples prepared by the routine diagnostic procedure (Figure 1). The cDNA was produced with either genotype specific primers (E590), or with both oligo-dT primers and random hexamers (E372, E026).

### 3.1. Sample E590

DRS reads from extracted RNA were obtained up to 7 h after the beginning of the sequencing run, and sequencing was stopped after 12 h. Out of a total output of 137,834 raw nanopore reads, 9213 reads were successfully basecalled into RNA sequences, with an average length of 1128 bases (range 1–7112 bases). More than 98% of basecalled RNA reads (9065 reads) were taxonomically classified using BLASTN against the NCBI’s nucleotide (*nt*) database. The large majority (>98%) of those hits matched eukaryotic sequences, and only 2.2% bacterial (199 reads) and 0.17% viral sequences (16 reads) (Table 1). The majority of eukaryotic reads were assigned to the yeast species *Saccharomyces cerevisiae* (8619 reads; 95%). All other eukaryotic species had less than five reads assigned. Within the reads assigned to bacteria, the only species with notable number of reads were *Escherichia coli* (37 reads), *Faecalibacterium prausnitzii* (12 reads) or *Bacteroides vulgatus* (12 reads), all of which are known to be commensal species in the human gut.

As far as viral sequences were concerned, all 16 (0.18% of 9065) reads matched the *Enterovirus* genus, species EV-A (Table 2). They were on average 2446 bases long (range 458–7112 bases). Alignment of the RNA sequences to the best-scoring reference genome (Coxsackievirus A6 KJ541158; Figure 2) demonstrated that the long reads covered almost the entire genome of Coxsackievirus A6, with one single read covering alone 98.5% of the reference genome (with only 110 bases missing at the 5′ end). The top two longest RNA sequences (7112 and 5397 bases long) were obtained within the first hour of sequencing, and most of the larger sequences (>2 kb) were obtained within the first 3 h of sequencing. After 3 h of sequencing, sequences matching enterovirus were all <900 bases long.

Classical amplicon sequencing of the VP1 region was attempted for sample E590, but it did not yield good results. In order to confirm genotype identification obtained via DRS and to obtain a highly accurate whole-genome enterovirus sequence, we also subjected sample E590 to cDNA sequencing using Illumina MiSeq. MiSeq sequencing library preparation was performed with stool material and following the routine diagnostic pre-treatment (as opposed to chloroform/bead treatment) given that low amount of patient stool sample was available (Figure 1). For this sample, an enterovirus-targeted approach was chosen to produce the cDNA by using genotype-specific primers given that low number of reads was obtained by DRS. After removal of reads mapping to human genome and bacterial rRNA, metatranscriptomic reads were assembled into contigs and further subjected to BLASTN similarity analysis against sequences in the *nt* database, and the top hits were taxonomically summarized using MEGAN. Most contigs of sample E590 were classified as bacteria (38) or viruses (17) (Table 1). Such differences in species composition when compared to the results of the DRS run may be explained by the genotype-specific approach used for cDNA synthesis and different pre-treatment of the sample.

All viral sequences matched again uniquely to EV sequences. However, MEGAN analyses revealed the presence of two enterovirus species consisting of EV-A (8 contigs; contigs lengths ranging from 130 to 3164 bases) and EV-B (8 contigs; contigs lengths ranging from 241 to 2,845 bases) in the cDNA sample, of which the majority of contigs were assigned to Coxsackievirus A6 and Echovirus 18 (Table 2). Mapping of the viral sequences to a VP1 database confirmed the presence of these two genotypes and the identification was verified by using the online RIVM bioinformatic platform [32]. Alignment of the reads to the closest reference genome sequences revealed that the coverage of CV-A6 was 96% with an average depth of 7.8 (±4.5), while for Echovirus 18, only 68% of the reference genome was covered (Figure 2). The consensus sequence for CV-A6 obtained by Illumina MiSeq was 94.0% identical (6926/7368), with 2% gaps (167/7368), to the consensus sequence obtained by DRS sequencing.

### 3.2. Sample E372

Based on the promising results obtained for sample E590, the DRS transcriptomic approach was repeated on two other clinical samples: For the DRS of RNA extracted from sample E372, the total yield of the sequencing run was 58,548 reads (14.84 Mb) after 12 h of sequencing. A total of 31,636 reads passed basecalling, with an average length of 144.4 bases (range of 2-4276). Only 42% (13,297) of the basecalled sequences were taxonomically classified using BLASTN. Interestingly, there was a drastic difference in read composition as compared to the first sample (E590), and the majority of reads classified as of bacterial origin (97.5%, 12,967), and only very few reads matched to eukaryotic (10, 0.8%) or archaeal (5, 0.4%) species. For bacteria, DRS reads were assigned to a large variety of species. The most prominent phyla were Bacteroidetes (4518 reads) and Proteobacteria (717 reads). A total of 315 viral reads, corresponding to 2.4% of classified reads, were found. As seen for the previous sample, only matches to the *Enterovirus* genus were found. Although many more viral reads were found in comparison to sample E590, they were, with an average of 1192 bases, much shorter, and with a range of 106–4276. No reads spanned the complete genome sequence of any known EV species or genotype, and altogether could cover 59% of the reference genome sequence (Figure 2). Short fragment lengths might result from increased fragmentation of the viral RNA during extraction or library preparation. Due to the directional sequencing used in the DRS process of nanopore sequencing, higher coverage at the 3’ end of the sequence may be obtained, and fragments without poly-A tail cannot be sequenced. Therefore fragmented, incomplete RNA molecules may hinder the recovery of the 5′ end of the RNA genome. Nevertheless, although the covered region did not span the VP1 region, we were able to identify the enterovirus as Echovirus 30 by genomic similarity to the best-scoring reference genome sequence. This species identification was independently confirmed by a standard genotyping approach that uses Sanger-based sequencing of the partial VP1 amplicon.

The cDNA from sample E372 was also sequenced by Illumina MiSeq, although in this case with non-specific primers for cDNA synthesis for a more unbiased approach towards EV detection and not to miss possible EV co-infections. Both oligo-dT and random hexamer primers were used to ensure a better chance of obtaining the full genome, with the idea that oligo-dT may lead to results more comparable to those of DRS due to polyadenylation requirements, and random hexamers may offer better chance of obtaining a well-covered 5’ end of the enterovirus genome. The distribution of contigs assigned on the domain level did not vary much between random hexamer and oligo-dT produced cDNA: most contigs were assigned to bacteria (99.3% or 99.5%, respectively), with only a small minority mapping to eukaryotes (0.15%, 0.20%) and viruses (0.51%, 0.28%). Within bacteria, most contigs were assigned to the phyla Proteobacteria (11,120, 12,551), followed by Bacteriodetes (735, 896) and Actinobacteria (303, 381). Taking a closer look at all the detected viral genotypes (Table 2), we found that, besides Echovirus 30, there were also contigs mapping to other virus genotypes, such as Rhinovirus A, and a variety of other bacterial or plant viruses, albeit only with few short contigs. The Echovirus 30 genome, however, was well covered with either approach (Figure 2). Comparison of the DRS nanopore sequencing consensus to the one obtained from Illumina MiSeq showed 97.3% identity (4211/4329), with 0.8% gaps (34/4329).

### 3.3. Sample E026

The DRS run of the RNA extract from sample E026 was stopped after the maximal run duration of 48 h. Notably, this run had much higher output compared to the other two samples, resulting in a total of 1.56 M sequenced reads (1.78 Gb), from which 1.46 M reads passed basecalling. The average length was also considerably higher than in previous runs with 1035 bases (range 1–7101 bases). Although the RNA input material used for library preparation was measured to be of a similar amount for all three samples (361 ng), cDNA measurement (896 ng) before loading on the flow cell indicated that the initial RNA concentration might not have been the same. Overall, 91% of basecalled reads (1.41 M reads) were taxonomically classified using BLASTN. Of those, 95.3% (1,349,407) mapped to eukaryotic sequences. As seen for sample E590, the vast majority of eukaryotic reads mapped to *Saccharomyces cerevisiae* (96.8%), while 4.6% (65,408) were classified as of bacterial origin, and only a negligible amount as archaea (6, 0.0004%). Viral sequences comprised 0.11% (1579) of the reads, the majority of which were classified as plant viral pathogens (Table 2), mostly belonging to the tomato mosaic virus (1274 reads, 81% of viral reads). Other, more rarely detected species, included Melon necrotic spot virus (8), and Cactus virus X (21). In total, 181 reads mapped to enteroviruses. Although sequences were mapped to several genotypes, mapping the reads to the best-scoring reference genome and to a VP1 database identified genotype Echovirus 25 as the most likely and only EV genotype present in the sample. Although there were relatively few hits to enteroviruses considering the high total output of the run, long sequences (average 1724.2, range 257–7101 bases) were observed, which covered up to 99.6% of the reference genome sequence (Figure 2).

Illumina sequencing of the cDNA from sample E026 produced using oligo-dT primers and random hexamers showed similar distributions on the domain level: the majority of contigs belonged to bacterial species (98.6% and 98.9%, respectively, for oligo-dT and random hexamer approaches), with a minority of reads mapping to eukaryotes (0.49%, 0.50%) and viruses (0.87%, 0.56%). This major shift from yeast to bacteria as compared to the DRS approach, which was also observed for sample E590, could be explained by the different pre-treatment using filtration with a pore size of 0.2 µm. Additionally, comparison of number of contigs and nanopore reads are of limited value when not considering coverage depth. For E026, the genome of Echovirus 25 genotype, whose presence was also confirmed by Sanger sequencing of the VP1 gene, was well covered with both types of cDNA synthesis (Figure 2). Some contigs also mapped to another genotype, Echovirus 30, but these were only short contigs, which may have been wrongly taxonomically assigned due to the short sizes of the Illumina reads. Otherwise, we also found a variety of viral sequences, with the most prominent being the tomato mosaic viruses and other plant pathogens, reflecting the results of the DRS run. The consensus sequence identity for Echovirus 25 sequenced by DRS to Illumina MiSeq was calculated as 96.7% (6979/7218), with 1% gaps (104/ 7218).

## 4. Discussion

Using RNA extracted from clinical samples, we were able to repeatedly identify human enteroviruses in stool samples from three independent patients by nanopore-based DRS in this proof-of-concept study. Beyond identifying EV species and genotypes correctly, we showed that the approach may also provide rich metatranscriptomic information on sample composition for all life domains. Clear differences in overall species compositions, with either yeast or bacteria dominating the majority of obtained reads, were observed between samples. Viral reads constituted between 0.11% and 2.4% of total passed reads. By complementing DRS data with Illumina MiSeq sequencing data for the same samples, we were able to validate the obtained enterovirus sequences and added further metatranscriptomic information. Illumina MiSeq sequencing revealed a higher diversity in viral species in these samples (Table 2), which was not captured by the DRS approach.

In all samples, 59% to 99.6 % of the EV reference genome sequences were covered. The average identity of the DRS consensus was 94–97% compared to the Illumina MiSeq consensus sequence. Other studies using DRS have achieved similar values with 93–97% identity when sequencing viruses from cell culture [20,33], or up to 99% consensus identity for influenza A [18]. For applications with samples with low target concentration such as patient samples, getting enough sequencing depth to obtain accurate consensus sequences constitutes a challenge for the method. The resulting consensus sequences might therefore be of limited use for applications which require high accuracy, such as phylogenomic analysis. However, for the purpose of genotype identification, the approach is sufficiently accurate, especially as it can provide long reads, which can facilitate mapping to the correct reference, and expedite downstream bioinformatic analyses. Furthermore, different EV genotypes have <75% nucleotide identity in the VP1 region [34,35], which makes VP1-based genotype identification possible even with the current high error rate of DRS (modal read accuracy of >90% [14]) when considering read length above few hundred bases.

Our wet laboratory approaches were refined during the course of the study while analyzing further samples, and the experiments had to be adapted to limited patient material availability. The good overall agreement in EV detection between the two NGS approaches suggests that the variability observed in number of reads and composition between samples may be attributable to the natural biological variability that may exist between the sampled patients. Yet, finer differences in composition between samples may be explained by the sequencing technology and by the different pre-treatments of the samples: on the one hand, smaller cDNA molecules produced by the wet laboratory procedure (cDNA synthesis, bead cleaning), followed by short-read sequencing, may be easier to detect, than larger, intact RNA molecules via DRS. On the other hand, unreliable mapping and poor taxonomical identification may be produced by short reads aligning to genomic regions that are conserved among genotypes, but also by long, error-prone nanopore reads [14] aligning incorrectly to reference sequences. Therefore, further large-scale comparisons of the two NGS approaches would be needed to confirm or reject the hypothesis about the effect of read length on organism detectability in metagenomic or metatranscriptomic studies.

While enteroviruses were detected in all tested samples using DRS, it should be noted that these were all samples with relatively high viral load, i.e., low Ct values. With a range of only 16–315 total reads mapping to enteroviruses for the three clinical samples, the approach was close to miss EV detection in those samples. In the case of sample E590, this sensitivity issue could explain why co-infection by another enterovirus was not detected by DRS, but only by the short-read sequencing-based approach. In a recent study on the use of DRS for identification of porcine reproductive and respiratory syndrome virus, viruses could be detected in spike-in samples with 3.4 × 10^4^ viral copies and in clinical samples with 3.8 × 10^6^ viral copies [33]. Another study has previously reported low sensitivity of nanopore sequencing in a viral metagenomic approach in patient samples with low viral titers, even when sequencing viral genomes via SISPA-based amplification [13]. An improvement in sensitivity would be necessary before attempting to sequence samples with low viral load in general. A limiting factor of the DRS approach was the amount of polyadenylated RNA used for library preparation, as in two of the sequencing runs, the full flowcell sequencing capacity was not used and the sequencing was stopped before maximal run duration, because low amounts of RNA library were loaded on the flowcell. If a sufficient amount of patient samples is available to obtain enough RNA, increasing the input concentration to fulfill the recommended input of at least 500 ng polyadenylated (as opposed to total) RNA might have provided better coverage of the target genome sequences. Furthermore, DRS sequencing is expensive if only one sample is loaded per flowcell. Recent progress with barcoding for DRS [36] might, however, reduce this limitation in the future. Additionally, results might be improved with adaptations in viral enrichment steps. We used a chloroform/bead pre-treatment in DRS samples for enrichment of viral capsid and although it might be efficient in enriching viral reads, it could also reduce the overall amount of RNA extracted. As the maximum sequencing yield of the flowcell was not always reached, perhaps milder enrichment methods might be preferable, especially if the broader diversity of pathogens is to be studied. Currently, the DRS method is restricted to sequencing of polyadenylated RNA. However, performing polyadenylation of the total RNA fraction could expand its applications.

Another characteristic of the DRS method is the expected sequencing bias towards the 3’ end of the polyadenylated molecules [14]. Hence, coverage of the 5’ region of the EV genomes was generally low, which may be problematic given that the VP1 region is the main region used for EV genotyping (e.g., located at position 2207–3082 in Echovirus 30 KT353720.1). Thus, correct genotyping would require around 4800 bases long RNA reads from the 3’ towards the 5’ end of the genome sequence. We encountered lesser coverage of the 5’ region of the genome with sample E372, although in that instance, we were still able to identify the same genotype as found by the gold-standard VP1 PCR-based approach. However, unambiguous identification relies on the capsid region and its presence is crucial in case of recombinations or co-infections. Therefore, great care must be applied during extraction and library preparation to avoid shearing or degrading extracted RNA. The known high variability of EV genome sequences makes it necessary to have suitable alternatives for genotype identification available if PCR-based approaches fail. In such cases, DRS of patient samples could be a valuable alternative, as the sequencing is primer independent and the sequences of long, native RNA molecules are recovered. Additionally, as is the case with all nanopore sequencing, the data is available in real time, which can provide faster identification and a sequencing-on-demand approach to molecular assays.

Overall, our proof-of-concept study demonstrated the possibilities offered by the DRS technique for genotype identification of human enteroviruses directly from clinical samples. The method requires further optimization to improve the overall sensitivity and to lower costs for possible applications in routine diagnostics. However, with the rapid advancement of the technologies, those issues will likely improve in the near future.

## Figures and Tables

**Figure 1 viruses-12-00841-f001:**
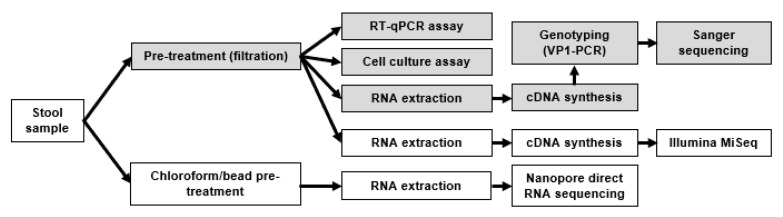
Analytical workflow. The workflow used for diagnostic assays is indicated by greyed boxes and that followed for NGS techniques by white boxes.

**Figure 2 viruses-12-00841-f002:**
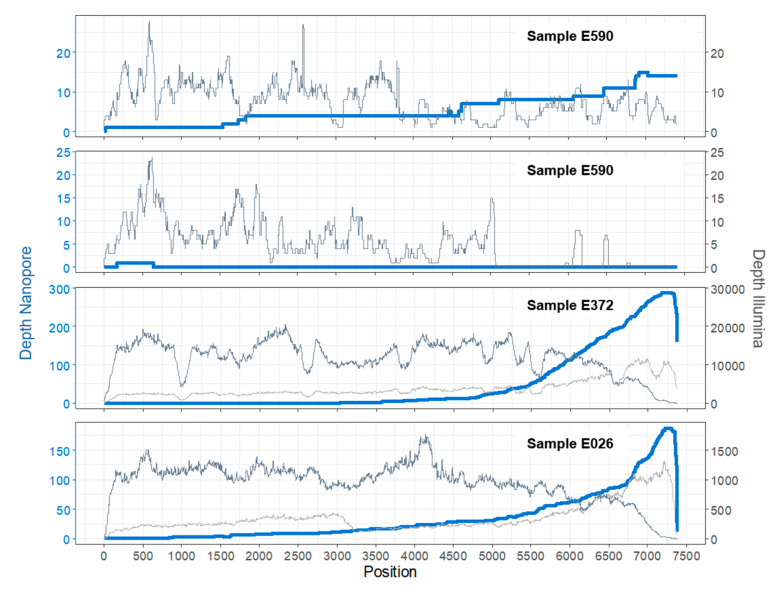
Coverage plots for both DRS nanopore runs (blue lines) and Illumina MiSeq (grey lines) for all three samples for each EV genotype. Shown are the depth of coverage after mapping the corresponding reads to the best reference genomes in sample E590 (top panel CA6: KJ541158.1, 2nd panel E18: HM777023.1) and E372 (MH484072.1). For E026, the coverage plot was produced by using the assembled contig sequence of the Illumina sequencing run (oligo-dT primers) as reads did not map well to any reference EV genome sequence. For samples E372 and E026, Illumina MiSeq reads based on cDNA produced with oligo-dT and random hexamers are indicated by light and dark grey lines, respectively.

**Table 1 viruses-12-00841-t001:** Taxonomic classification of reads (Oxford Nanopore) and contigs (Illumina MiSeq). BLASTN matches against the NCBI’s nucleotide (*nt*) database were summarized at the domain level with MEGAN.

	Sample E590		Sample E372			Sample E026		
Domain	Nanopore (reads)	Illumina MiSeq (contigs)	Nanopore (reads)	Illumina MiSeq (contigs)		Nanopore (reads)	Illumina MiSeq (contigs)	
	DRS	Genotype specific primers	DRS	Random hexamers	Oligo-dT primer	DRS	Random hexamers	Oligo-dT primer
**Eukaryota**	8850	1	10	19	28	1,349,407	227	56
**Bacteria**	199	38	12,967	12,411	14,125	65,408	45,096	11,004
**Archaea**	0	0	5	1	0	6	7	3
**Viruses**	16	17	315	64	40	1579	398	63
**Total**	9065	56	13,297	12,495	14,193	1,350,992	45,728	11,126

**Table 2 viruses-12-00841-t002:** Taxonomic classification of viral reads (Oxford Nanopore) and contigs (Illumina MiSeq) using MEGAN based on BLASTN matches against the NCBI’s nucleotide (*nt*) database at the species and genotype levels. Asterisks (*) indicate that >90% of the respective genome sequence was covered, while bold numbers indicate EV species identity confirmed by Sanger sequencing of the VP1 gene. Minus sign (−) indicates absence of hits in the dataset.

	Sample E590		Sample E372			Sample E026		
Run ID	E590-DRS	E590-MiSeq	E372-DRS	E372-MiSeqR6	E372-MiSeqOdT	E026-DRS	E026-MiSeqR6	E026-MiSeqOdT
Technology	DRS	MiSeq (custom primers)	DRS	MiSeq (random hexamers)	MiSeq (oligo dT)	DRS	MiSeq (random hexamers)	MiSeq (oligo dT)
Coxsackievirus A6	16*	8	−	−	−	−	−	−
Coxsackievirus A9	−	−	−	−	−	1	−	−
Coxsackievirus B1	−	−	−	−	−	4	−	−
Coxsackievirus B2	−	−	1	−	−	−	−	−
Coxsackievirus B3	−	1	1	−	−	5	−	−
Coxsackievirus B5	−	−	1	−	−	1	−	−
Echovirus 18	−	4	−	−	−	−	−	−
Echovirus 25	−	−	−	−	−	**12 ***	**1 ***	**1 ***
Echovirus 30	−	−	**124**	**28**	**1 ***	−	9	2
Echovirus 7	−	−	6	−	−	−	−	−
Echovirus E12	−	−	−	−	−	5	−	−
Echovirus E18	−	−	−	−	−	1	−	−
Echovirus E9	−	−	−	−	−	1	−	−
Enterovirus B106	−	−	−	−	−	2	−	−
Enterovirus B80	−	−	−	−	−	1	−	−
Enterovirus B81	−	−	−	−	−	5	−	−
Enterovirus B85	−	−	−	−	−	2	−	−
Enterovirus B86	−	−	−	−	−	1	−	−
Enterovirus B88	−	−	−	−	−	4	−	−
Rhinovirus A	−	−	−	7	8	−	−	−
Bovine arvovirus 3	−	−	−	−	1	−	−	−
Cactus virus X	−	−	−	−	−	21	12 *	5
Caprine arthritis encephalitis virus	−	−	−	−	−	−	1	−
Carrot cryptic virus	−	−	−	−	−	−	4	1
Caudovirales	−	−	−	3	2	−	22	5
Escherichia virus FI	−	−	−	−	−	−	1 *	2
Escherichia virus phiX174	−	−	−	1 *	1	−	2	1
Gokushovirus WZ−2015a	−	−	−	−	−	−	−	1
Human gut gokushovirus	−	−	−	−	−	−	−	1
Melon necrotic spot virus	−	−	−	1	−	8	1 *	1 *
Pitaya virus X	−	−	−	−	−	−	4	−
Schlumbergera virus X	−	−	−	−	−	−	12	−
Tomato mosaic virus	−	−	−	2	5	1274 *	248	1 *
Tomato mottle mosaic virus	−	−	−	−	−	−	5	1
Unclass. bacterial viruses	−	−	−	18	21	−	61	37
Uncultured Microviridae	−	−	−	−	−	−	1	1

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
