# Peer review of "Whole-Genome Sequencing of Human Enteroviruses from Clinical Samples by Nanopore Direct RNA Sequencing"

_viruses, 2020, doi:10.3390/v12080841_

Round 1
Reviewer 1 Report
Abstract:
L13: in place of the word "those", perhaps use "enteroviruses".
L39: in place of "sequencing short sequences alone", perhaps use " sequencing single genomic regions"
Methods:
L96/L120: Was the Trizol LS method also used for DRS? please clarify.
L133/L140/L192: Why were Coxsackievirus A6 primers used with E590 sample only? and not the other two samples? The authors should clarify this in the manuscript.
L153: Similarly, why was there an additional end-prep step for E590?
L158: And use of a different library prep kit for E590?
L170: How was the 'best scoring reference genome' selected or determined?
L178: Perhaps rephrase as "Coverage information was generated using BBMap and plots created using R ..."
Results:
L220 to L223: The statement start with "In this case ... (Figure 1)" is not clear. Please re-phrase it. On the same statement, what do authors refer to by "original material"? is that the stool sample or the RNA?
L233: By "majority of reads", did you mean "contigs"?
L255: rectify: any known EV species ..
L291: rectify: "compared to the other two samples"
L307: provide average and range of the sequences refered to in ".. some very long sequences .."
Discussion:
L325 to 327: This sentence is not clear.
L335: similarly, what do the authors mean by ".. and the further analysis of metatranscriptomic data"
L362: what do the authors mean by ".. the approach was yet close to the limit of detection"; what limit of detection do the authors refer to?
L368: "A limiting factor of the DRS approach .."
L371: "If sufficient amounts .." - what do the authors think might be sufficient? based on the amount used in the study? It will be useful if this could be mentioned or speculated for readers to know
L385-387: Based on authors knowledge or other published studies, what would be the minimum sequence length (threshold) for genotyping based on VP1? what would be the threshold of sequence diversity below which you cannot differentiate between various EV genotypes based on VP1?
Author Response
Abstract:
L13: in place of the word "those", perhaps use "enteroviruses".
Authors' response: We have changed the text accordingly.
L39: in place of "sequencing short sequences alone", perhaps use " sequencing single genomic regions"
Authors' response: We have changed the text accordingly.
Methods:
L96/L120: Was the Trizol LS method also used for DRS? please clarify.
Authors' response: The Trizol LS extraction method was used for all DRS and Illumina extractions except the DRS sequencing of sample E590. This was changed as part of improving the protocol. All this information is also available in detail in supplementary table 1, which we refer to on lines 203/204 in the results section. We have also clarified this now in the materials and methods section on L124-126.
L133/L140/L192: Why were Coxsackievirus A6 primers used with E590 sample only? and not the other two samples? The authors should clarify this in the manuscript.
Authors' response: We have improved the protocol in the course of the experiments, as indicated in the text. In the first experiment, we chose an enterovirus specific approach, but given that a co-infection was found, we decided to use unspecific, and more general approach for subsequent experiments. We have modified some sentences in the Results sections, lines 245-248, to further clarify the issue. It is also mentioned in lines 303-305.
L153: Similarly, why was there an additional end-prep step for E590?
Authors' response: As mentioned in the previous comment, this was also part of an improvement in the protocol as this was an unnecessary step for the Illumina based approach. We have added another sentence on line 165 to clarify this point.
L158: And use of a different library prep kit for E590?
Authors' response: The library kit was also changed during protocol optimization. We have modified line 172 as well to reflect this point.
L170: How was the 'best scoring reference genome' selected or determined?
Authors' response: The best scoring reference genome was selected by choosing the whole genome sequence with the highest bit score in the BLASTn output. (added to line 189/190).
L178: Perhaps rephrase as "Coverage information was generated using BBMap and plots created using R ..."
Authors' response: Rephrased as suggested.
Results:
L220 to L223: The statement start with "In this case ... (Figure 1)" is not clear. Please re-phrase it. On the same statement, what do authors refer to by "original material"? is that the stool sample or the RNA?
Authors' response: It refers to the patient stool sample material. The sentence has been divided and rephrased for clarification. (lines 245-248)
L233: By "majority of reads", did you mean "contigs"?
Authors' response: That is correct, we have changed it in the text.
L255: rectify: any known EV species ..
Authors' response: Done accordingly.
L291: rectify: "compared to the other two samples"
Authors' response: Modified accordingly.
L307: provide average and range of the sequences refered to in ".. some very long sequences .."
Authors' response: The information was added to the text.
Discussion:
L325 to 327: This sentence is not clear.
Authors' response: The sentence refers to the changes in protocol that were done during the course of the study. The sentence has been rewritten and moved to lines 386/387.
L335: similarly, what do the authors mean by ".. and the further analysis of metatranscriptomic data"
Authors' response: This statement refers to metatranscriptomic data that was obtained by Illumina MiSeq, to complement the results from DRS. We have changed the sentence to clarify its meaning (lines 367/368).
L362: what do the authors mean by ".. the approach was yet close to the limit of detection"; what limit of detection do the authors refer to?
Authors' response: We refer to the limit of detection of EV by the DRS method. It was meant that the enterovirus genomes were in some cases close to being undetected, given that were only 16 reads available and it might not be reproductible if only very few reads are detected. The formulation was changed in the text.
L368: "A limiting factor of the DRS approach .."
Authors' response: Modified as suggested.
L371: "If sufficient amounts .." - what do the authors think might be sufficient? based on the amount used in the study? It will be useful if this could be mentioned or speculated for readers to know
Authors' response: The recommended nanopore input concentration is currently of 500 ng polyA RNA. While we had RNA concentrations in that range, it refers to total RNA, not only polyadenylated RNA. In addition, our samples contained mRNA from non-viral organisms, which also contributes to non-target RNA reads. We feel that it is too early to provide recommendation about sufficient amounts given that other factors may also induce unknown fluctuation in the protocol such as library preparation efficiency, which is not quantifiable at this stage. Therefore, we cannot speculate too much about the sufficient amount for either successful metatranscriptomics or detection of RNA virus application. Those points are clarified in the text (lines 416/417).
L385-387: Based on authors knowledge or other published studies, what would be the minimum sequence length (threshold) for genotyping based on VP1?
Authors' response: Genotyping is also considered valid with partial sequencing of VP1 of a length of around 375 bp (Nix et al. 2006), which is located for example at position 2,291-2,668 of the genome Echovirus 30 KT353720.1. If this region is covered, VP1 based genotyping is possible.
what would be the threshold of sequence diversity below which you cannot differentiate between various EV genotypes based on VP1?
Authors' response: EV must have <75% nucleotide identity in the VP1 region to be classified as a different genotype (Oberste et al. 1999, Brown et al. 2009). With this high diversity between genotypes, we believe VP1-based genotype assignment should be possible even with the high error rate of DRS (modal read accuracy of >90% (Garalde et al. 2018)) when considering read length of above a few hundred bp. We have added this information in the text on lines 381-384.
Reviewer 2 Report
Small scale study on the potential benefits of using DRS for routine diagnosis.
Although a small number of samples were used, the results obtained were critically approached stating both benefits and pitfalls of the technique.
Author Response
Authors' response: Thank you.
Reviewer 3 Report
In the manuscript “Whole genome sequencing of human enteroviruses from clinical samples by nanopore direct RNA sequencing”, Grädel and co-authors evaluate the adoption of Oxford Nanopore’s direct RNA sequencing to genotype enteroviruses from stool samples and to retrieve their whole genome sequence. Even considering the need for further optimization of the protocols, as pointed out by the authors in the conclusions section, the method would be a good option for sequencing and genotyping enteroviruses in the future.
The findings are well supported by the data and the results are well presented. The manuscript is generally well written, with only a few sections that could benefit from a more detailed description.
I recommend accepting this manuscript for publication pending minor revisions.
I only have a few minor comments / suggestions:
1) I would consider citing by Tan et al., 2019 (doi 10.3390/v11121132), where they describe Nanopore’s direct RNA sequencing for strain detection of porcine reproductive and respiratory syndrome virus (PRRSV) and investigate analytical sensitivity and the error-rate of Nanopore’s DRS.
2) lines 163-167: Did you trim the adapters from Nanopore’s DRS reads? If yes, which tool did you use? I see that you used BBduk to trim adapters from Illumina reads, but I could not understand if the adapters in Nanopore reads have been trimmed.
3) further to comment no.2, if Nanopore adapters have not been trimmed, could the low percentage of reads classified with Blastn be due to presence of the adapters sequences, or to the presence of chimeric reads (which could be identified, having internal adapters)?
4) lines 169 and 176-177: Did you filter the results of Blastn search against the NCBI nt database (e.g. only considering matches with E-value lower than a certain threshold), or you just considered the first match for each read/contig, regardless of the E-Value? Please explain in the materials and methods.
5) I suppose each library for Nanopore direct RNA sequencing was loaded on a different flow cell. Is it correct? Or you have washed the flow cell and re-used it for the next sample? Please describe in the materials and methods.
5) line 196 and supplementary table 2: In the manuscript you mention a total output of 102,391 raw reads from DRS of sample E590. In supplementary table 2, the total (raw) reads for sample E590 are 137,834. Is this a mistype?
6) lines 260-262 and table 2: In table 2 you list, for sample E372, reads classified as Coxsackievirus B1, B2, B3, B5 and Echovirus 7. There is no mention of these in the text. Maybe the reads were too short to be classified with confidence? Or the E-Value of the Blastn alignment was too high and you did not consider this reads? Please explain in the text.
7) lines 304-306 and table 2. Please explain with more details why you concluded that, based on DRS reads, Echovirus 25 is the most likely and only EV genotype in sample E026.
Author Response
In the manuscript “Whole genome sequencing of human enteroviruses from clinical samples by nanopore direct RNA sequencing”, Grädel and co-authors evaluate the adoption of Oxford Nanopore’s direct RNA sequencing to genotype enteroviruses from stool samples and to retrieve their whole genome sequence. Even considering the need for further optimization of the protocols, as pointed out by the authors in the conclusions section, the method would be a good option for sequencing and genotyping enteroviruses in the future.
The findings are well supported by the data and the results are well presented. The manuscript is generally well written, with only a few sections that could benefit from a more detailed description.
I recommend accepting this manuscript for publication pending minor revisions.
Authors' response: Thank you for your positive feedback.
I only have a few minor comments / suggestions:
1) I would consider citing by Tan et al., 2019 (doi 10.3390/v11121132), where they describe Nanopore’s direct RNA sequencing for strain detection of porcine reproductive and respiratory syndrome virus (PRRSV) and investigate analytical sensitivity and the error-rate of Nanopore’s DRS.
Authors' response: We have now added a citation for the suggested paper on L375 (for the accuracy) and refer to it in more detail concerning DRS sensitivity on L406-409.
2) lines 163-167: Did you trim the adapters from Nanopore’s DRS reads? If yes, which tool did you use? I see that you used BBduk to trim adapters from Illumina reads, but I could not understand if the adapters in Nanopore reads have been trimmed.
Authors' response: We have not trimmed the nanopore DRS reads for two reasons. 1) The adapter added to the RNA is a DNA molecule, which the RNA basecaller is not able to accurately basecall and is therefore difficult to identify (Viehweger et al. 2019). To our knowledge, there is currently no accurate way of removing the sequencing adapter for DRS and other studies have also not removed them. 2) Our downstream use of the DRS reads is to BLAST the reads to nt database. Because BLAST is a local aligner, the presence of non-matching sequences at the extremities of the reads is not problematic to identify the most closely related reference sequences in the databases. We have added this information in the methods section (lines 182-184).
3) further to comment no.2, if Nanopore adapters have not been trimmed, could the low percentage of reads classified with Blastn be due to presence of the adapters sequences, or to the presence of chimeric reads (which could be identified, having internal adapters)?
Authors' response: The length of the adapter is 30 bp and "short" RNA nanopore reads are typically in the range of few hundred bases, so it is unlikely that any known transcripts, even if harboring sequencing adapters, will not be correctly identified via BLAST similarity analysis. For sure, we cannot exclude the potential presence of chimeric reads, which we have not examined at this stage.
4) lines 169 and 176-177: Did you filter the results of Blastn search against the NCBI nt database (e.g. only considering matches with E-value lower than a certain threshold), or you just considered the first match for each read/contig, regardless of the E-Value? Please explain in the materials and methods.
Authors' response: The filtering and taxonomic analysis of the BLAST results were done using MEGAN with the following parameters for the program blast2rma: -c false \ -m 10 -ms 50 -me 0.01 -mpi 0 -top 10 -supp 0.05 -sup 1 -alg naive \ -mrc 0 -mrefc 0 -ram readCount \ -a2t PATH/nucl_acc2tax-Jul2019.abin. Those parameters are now indicated in the Methods section (lines 187/188).
5) I suppose each library for Nanopore direct RNA sequencing was loaded on a different flow cell. Is it correct? Or you have washed the flow cell and re-used it for the next sample? Please describe in the materials and methods.
Authors' response: Yes, that is correct. It is now described in the materials and methods on line 141.
5) line 196 and supplementary table 2: In the manuscript you mention a total output of 102,391 raw reads from DRS of sample E590. In supplementary table 2, the total (raw) reads for sample E590 are 137,834. Is this a mistype?
Authors' response: That was a typo in the text, thank you for pointing it out. We have corrected the text accordingly.
6) lines 260-262 and table 2: In table 2 you list, for sample E372, reads classified as Coxsackievirus B1, B2, B3, B5 and Echovirus 7. There is no mention of these in the text. Maybe the reads were too short to be classified with confidence? Or the E-Value of the Blastn alignment was too high and you did not consider this reads? Please explain in the text.
Authors' response: We did not mention the information in the text because we avoided presenting the same information in both Table 2 and in the text. We also focused only on the genotypes that were clearly shown to be present via VP1-based genotyping, as both sequencing technologies produced some non-target hits due to the sequencing of conserved regions across genotypes, which was mostly observed when using Illumina MiSeq approach (given the shorter reads produced). As reported in Table 2, for DRS, the other non-target EV hits never recovered the complete genome sequences. These non-specific hits were mentioned in Table 2 and also discussed in the Discussion section, but not in the Results section.
7) lines 304-306 and table 2. Please explain with more details why you concluded that, based on DRS reads, Echovirus 25 is the most likely and only EV genotype in sample E026.
Authors' response: Again, it is dependent on genome coverage provided by the DRS, and on the additional information obtained via VP1-based genotyping, as indicated in Table 2 and in the text.